# Facile In Situ Synthesis of Co(OH)_2_–Ni_3_S_2_ Nanowires on Ni Foam for Use in High-Energy-Density Supercapacitors

**DOI:** 10.3390/nano12010034

**Published:** 2021-12-23

**Authors:** Xuan Liang Wang, En Mei Jin, Jiasheng Chen, Parthasarathi Bandyopadhyay, Bo Jin, Sang Mun Jeong

**Affiliations:** 1Department of Chemical Engineering, Chungbuk National University, 1 Chungdae-ro, Seowon-gu, Cheongju 28644, Chungbuk, Korea; wangxuanleon@gmail.com (X.L.W.); jinenmei@chungbuk.ac.kr (E.M.J.); chenjs3824@gmail.com (J.C.); partha012@gmail.com (P.B.); 2Key Laboratory of Automobile Materials, Ministry of Education, and College of Materials Science and Engineering, Jilin University, Changchun 130022, China; jinbo@jlu.edu.cn

**Keywords:** nickel sulfide, cobalt hydroxides, core–shell, high energy density, asymmetric supercapacitor

## Abstract

Ni_3_S_2_ nanowires were synthesized in situ using a one-pot hydrothermal reaction on Ni foam (NF) for use in supercapacitors as a positive electrode, and various contents (0.3−0.6 mmol) of Co(OH)_2_ shells were coated onto the surfaces of the Ni_3_S_2_ nanowire cores to improve the electrochemical properties. The Ni_3_S_2_ nanowires were uniformly formed on the smooth NF surface, and the Co(OH)_2_ shell was formed on the Ni_3_S_2_ nanowire surface. By direct NF participation as a reactant without adding any other Ni source, Ni_3_S_2_ was formed more closely to the NF surface, and the Co(OH)_2_ shell suppressed the loss of active material during charging–discharging, yielding excellent electrochemical properties. The Co(OH)_2_–Ni_3_S_2_/Ni electrode produced using 0.5 mmol Co(OH)_2_ (Co_0.5_–Ni_3_S_2_/Ni) exhibited a high specific capacitance of 1837 F g^−1^ (16.07 F cm^−2^) at a current density of 5 mA cm^−2^, and maintained a capacitance of 583 F g^−1^ (16.07 F cm^−2^) at a much higher current density of 50 mA cm^−2^. An asymmetric supercapacitor (ASC) with Co(OH)_2_–Ni_3_S_2_ and active carbon displayed a high-power density of 1036 kW kg^−1^ at an energy density of 43 W h kg^−1^ with good cycling stability, indicating its suitability for use in energy storage applications. Thus, the newly developed core–shell structure, Co(OH)_2_–Ni_3_S_2_, was shown to be efficient at improving the electrochemical performance.

## 1. Introduction

Currently, the main concerns in society are efficient energy storage and clean energy. Supercapacitors are the most ideal devices for use in clean energy storage, owing to their low costs, rapid discharge rates, and high electrochemical activities, in addition to mechanical and thermal stabilities [1,2]. However, despite numerous advantages, supercapacitors exhibit lower energy densities (~10 W h kg^−1^) compared to those of Li-ion batteries (LIBs, typically <300 Wh kg^−1^) [3,4]. Supercapacitors can be divided into electrochemical double-layer capacitors (EDLCs) and pseudocapacitors. EDLCs employ an electric double layer formed at the interface of an electrolyte and a polarizable electrode, such as activated carbon (AC), and pseudocapacitors are based on the Faradaic electrochemical redox mechanism of charge storage [5]. Carbons with high surface areas are mainly used in EDLCs, whereas in pseudocapacitor research, oxides, such as ruthenium and manganese, are mainly used. According to a review of recent advances in carbon–based materials for use in supercapacitor electrodes, the specific energy is generally <10 Wh kg^−1^ [3,6,7]. Recently, mono– and bimetallic oxides, layered double hydroxides, and transition metal sulfides were proposed to overcome the lower energy densities in supercapacitors [8,9,10,11]. The transition metal sulfides (nickel or cobalt sulfide) exhibited considerable potential; in particular, nickel sulfides exhibit excellent electrochemical energy storage and fascinating properties, such as excellent redox reversibility, conductivities, and capacitances [12,13,14,15]. However, low-rate capability and poor electrochemical stability limit practical applications because of the surface Faradaic redox reactions and reverse kinetics during rapid charge−discharge [16]. Nickel sulfides have various phases, such as α– and β–NiS, NiS_2_, and Ni_3_S_2_ are low−cost, abundant materials with widespread applications, such as LIB and supercapacitor electrode materials [17,18,19,20].

The most recently reported nickel sulfide-based research mainly focuses on engineering vacancies including control of morphologies and phase crystallinities, binding them with other carbon materials, and doping to optimize electrochemical performance [21,22,23,24,25]. For example, Chou et al. prepared a binder-free flaky Ni_3_S_2_ electrode as a positive material that displayed a high specific capacitance (SC) of 717 F g^−1^ at 2 A g^−1^ and excellent cycle retention of 91% [24]. Krishnamoorthy et al. used a one-pot hydrothermal method to prepare nest–type Ni_3_S_2_ with an SC of 1293 F g^−1^ at 5 mA cm^−2^ [25]. Li et al. used single–crystal β–NiS nanorod arrays to synthesize Ni_3_S_2_@β−NiS materials with novel 3D architectures and pine twig-like morphologies in hollow Ni_3_S_2_ porous frameworks, with an SC of 1158 F g^−1^ at 2 A g^−1^ [26]. Hu et al. proposed a phase modulation strategy dominated by a novel coordination agent supported by a hydrothermal process [27].

In addition, the crystallization and morphology of nickel sulfide are primarily tuned by controlling the sulfur source in the reaction. In general, nickel sulfide is synthesized by dissolving metal salts, such as nickel acetate, nitrate, and chloride, and sulfur precursors, such as sodium sulfate, thioacetamide (TAA), thiourea (TU), and sodium thiosulfate, in solution [28,29,30,31,32,33,34]. Chen et al. used Ni(NO_3_)_2_·6H_2_O and TU to directly prepare Ni_3_S_2_ on Ni foam (NF) [35], and Wu et al. used TAA and NF to prepare rod–like Ni_3_S_2_ [36].

In addition, numerous studies were conducted to evaluate core–shell structures to improve the electrochemical properties of Ni_3_S_2_-based supercapacitors [37,38,39,40]. The core–shell structure protects the active material, ensuring excellent cycle stability during charging and discharging. However, multi-step reactions are generally performed to prepare core–shell structures. For example, Chen et al. used a two-step hydrothermal reaction to fabricate the core–shell structures of Ni_3_S_2_@Co(OH)_2_ nanowires formed directly on NF for use in asymmetric supercapacitors (ASCs), and the Ni_3_S_2_@Co(OH)_2_ electrode exhibiting a high SC of 2139.4 F g^−1^ at 2 mA cm^−2^, and maintaining a capacitance of 1139.4 F g^−1^ at 40 mA cm^−2^ [37].

In this study, the core–shell structured Co(OH)_2_−Ni_3_S_2_ was synthesized in situ using a one–step hydrothermal reaction on NF as a binder-free positive electrode for use in supercapacitors. By direct NF participation as a reactant, without adding any other Ni source, Ni_3_S_2_ was formed more closely to the NF surface, and the Co(OH)_2_ shell suppressed the loss of active material during charging–discharging. Moreover, NF provides sufficient microcavities for simple penetration of electrolyte ions and rapid transport of the generated gas. To suppress the active material loss during cycling, various amounts of Co were added to control the thickness of the shell, yielding excellent electrochemical properties. In addition, the electrochemical properties were investigated by assembling an ASC using Co(OH)_2_−Ni_3_S_2_ and AC as the positive electrode and negative electrode, respectively.

## 2. Experimental

### 2.1. Materials

KOH (95%), sulfur powder, and CoCl_2_·6H_2_O (98%) were obtained from Sigma–Aldrich (St. Louis, MO, USA). For ethyl alcohol (anhydrous, 99%), SAMCHUN chemicals (Gyeonggi-do, Republic of Korea) products were used, and for nickel form (NF, 99.5%), Goodfellow (Huntingdon, UK) products with a thickness of 1.5 mm and a bulk density of 0.45 g/cm^3^ were used.

### 2.2. Synthesis of Ni_3_S_2_ and Co(OH)_2_–Ni_3_S_2_ Positive Electrode

Ni_3_S_2_ nanowires for use as positive active materials were synthesized using a one-pot hydrothermal reaction. For synthesizing the Ni_3_S_2_ nanowires, the NF (2 × 1 cm^2^) was used as the reactant and substrate. First, NF was sonicated in 3 M HCl for 30 min, and then washed for 30 min by deionized water and ethanol with volume ration of 1:1 to remove organic impurities and surface oxidized layer. The washed NF was subsequently vacuum-dried overnight at 50 °C.

The detailed experimental procedure for synthesizing the NF–supported Ni_3_S_2_ nanowires is shown in Figure 1. The source of sulfur was the natural sulfur powder in molar ratios of 2 mmol. Sulfur powder (99.5%) was added to 8 mL of 3 M KOH and sonicated at 70 °C until the solution turned transparent yellow. A total of 8 mL of ethyl alcohol and NF (2 × 1 cm^2^) were then added, and the solution was sonicated for a few minutes. Subsequently, the uniform solution was transferred to hydrothermal reactor. The reactor was heated for 18 h at 160 °C. The product, Ni_3_S_2_, was collected after the reaction system was cooled to below 30 °C. Thereafter, the Ni_3_S_2_ film was washed and vacuum–dried at 50 °C overnight. During the hydrothermal process, sulfur reacts with the NF to uniformly deposit Ni_3_S_2_ on the NF surface. The reaction formula for the formation of nickel sulfide on the NF may be as below:3OH^−^ + 4S → 2S^2−^ + S_2_O_3_^2−^ + 3H^+^
Ni + 2OH^−^ → Ni(OH)_2_ + 2e^−^

3Ni(OH)_2_ + 2S^2−^ → Ni_3_S_2_ + 6OH^−^


For preparation of core–shell structured Co(OH)_2_–Ni_3_S_2_, after all sonication steps were completed, the 0.3, 0.4, 0.5, or 0.6 mmol of the source of cobalt was added and in the solution, followed by hydrothermal reaction at 160 °C for 18 h. The 0.3, 0.4, 0.5, or 0.6 mmol cobalt–containing Co(OH)_2_–Ni_3_S_2_ electrodes are denoted as Co_0.3_–, Co_0.4_–, Co_0.5_–, and Co_0.6_–Ni_3_S_2_, with a mass loading of 6.2, 6.8, 7.3, and 8.1 mg cm^−^^2^, respectively.

### 2.3. Cell Assembly

For the electrochemical performance study, a three-electrode system was assembled using material formed in situ on NF (area ≈ 1 × 1 cm^2^). Ni_3_S_2_ or Co(OH)_2_–Ni_3_S_2_ was used as the working electrode and Ag/AgCl and a Pt plate were used as the reference and counter electrodes, respectively. In the three-electrode configuration, 3 M KOH solution was used as the electrolyte.

The ASC device was assembled by Co_0.5_–Ni_3_S_2_ and AC electrodes as the positive and as negative electrodes, respectively. KOH (3 M) and filter paper (Whatman 42) were used as the electrolyte and separator, respectively. Prior to assembling the ASC device, the AC negative electrode was prepared as follows: super-p carbon black, polyvinylidene fluoride, and AC as the conductive material, binder, and active material, respectively, were mixed in a mass ratio of 1:1:8 in *N*–methyl–2–pyrrolidinone (NMP) to form a uniform slurry. The slurry was uniformly coated on a Ni mesh and dried in an electrical oven at 60 °C for 1 h. The dried electrode was pressed and vacuum–dried at 110 °C for a day. In addition, the filter paper and as–prepared Co_0.5_–Ni_3_S_2_ were immersed in electrolyte for a day. The ASC device was then assembled by sequentially stacking the soaked Co_0.5_–Ni_3_S_2_ electrode, filter paper, and AC electrode.

### 2.4. Materials Characterization and Electrochemical Measurements

The crystal structures and morphologies were characterized using X-ray diffraction (XRD, Ultima IV, Rigaku, Tokyo, Japan), field emission scanning electron microscopy (FE-SEM, LEO–1530, Carl Zeiss, Oberkochen, Germany), and FE transmission electron microscopy (FE–TEM, 200 KV, JEM–2100F, JEOL, Tokyo, Japan). To evaluate the elemental composition and oxidation state, X-ray photoelectron spectroscopy (XPS) was performed using a PHI Quantera–II (Ulvac–PHI, Chigasaki, Japan).

The electrochemical performance of the three-electrode configuration was measured using an Autolab electrochemical workstation (AUT84455, Metrohm, Herisau, Switzerland). Electrochemical impedance spectroscopy (EIS) was performed in the frequency range 10^5^–10^−2^ Hz at an amplitude of 10 mV using an Autolab electrochemical workstation (PGSTAT302N, Metrohm, The Netherlands). All resistances were determined using the NOVA program (Version 1.10.4, Metrohm, The Netherlands) to fit the related EI spectra. Cyclic voltammetry (CV, WBCS3000L, WonATech, Seoul, Korea) was measured at a scanning rate of 0.1 mV s^−1^ in the voltage range 1.5–3.2 V, and charge–discharge studies were performed at 0.1 C with a cut-off voltage of 1.5–3.2 V.

## 3. Results and Discussion

The XRD patterns of the Ni_3_S_2_ and Co(OH)_2_–Ni_3_S_2_ electrodes are shown in Figure 2. The XRD patterns of all samples are consistent with the hexagonal phase (JCPDS card no. 44–1418) in the R32(155) space group, with sharp, intense peaks at 2θ = 21.75°, 31.10°, 37.78°, 44.33°, 49.73°, 50.12°, 55.16°, and 55.34° representing the (101), (110), (003), (202), (113), (211), (122), and (300) diffraction planes, respectively [41]. Peaks representing the cubic phases corresponding to nickel (JCPDS card no. 87–0712) are observed at 2θ = 44.50° (111), 51.85° (200), and 76.38° (220) [42]. These nickel patterns are observed because Ni_3_S_2_ was formed on the NF. The Co(OH)_2_–Ni_3_S_2_ electrodes exhibit hexagonal and cubic phases, and their patterns lack the characteristic peaks representing Co(OH)_2_ because they overlap with the hexagonal peaks representing Co(OH)_2_ and/or the amorphous phase, which is formed by the hydrothermal reaction after CoCl_2_ addition [43]. Peaks representing Co(OH)_2_ corresponding to the hexagonal phase (JCPDS card no. 45–0031) are typically observed at 2θ = 32.22° (100), 37.65° (101), 51.09° (102), and 57.63° (110) [44].

Figure 3 shows the surface morphologies of the Ni_3_S_2_ and Co(OH)_2_–Ni_3_S_2_ electrodes with different Co contents were characterized using FE-SEM and elemental mapping analysis. Ni_3_S_2_ exhibits a smooth surface, with the Ni_3_S_2_ nanowires uniformly formed on the NF surface following the hydrothermal reaction, clearly indicating that the NF reacts with the added sulfur. The Ni_3_S_2_ nanowires form very uniformly in terms of length and thickness, and no other form of Ni_3_S_2_ is observed in the SEM images. As shown in the highly magnified SEM image of Ni_3_S_2_ (inset of Figure 3b), long, thin nanowires are observed, with an average diameter of ~150 ± 10 nm. As the amount of added Co increases, the smoothness of the nanowire surface decreases and the Co(OH)_2_ shell thickness increases (Figure 3c–f), because Co(OH)_2_ is only formed on the surfaces of the Ni_3_S_2_ nanowires, rendering their surfaces rough. A rough surface may exhibit high electrolyte adsorption capacity due to its large specific surface area, and Co(OH)_2_ may suppress the Ni_3_S_2_ active material loss during the cycling, resulting in excellent electrochemical properties. In addition, as shown by the side views of the elemental mapping images (Figure 3g,h), all elements are distributed uniformly on the nanowires. Therefore, the synthesis via hydrothermal reaction at 160 °C for 18 h is successful.

HRTEM was conducted to further explore the structures of the Co_0.5_–Ni_3_S_2_ nanowires. As shown in Figure 4a,b, the Ni_3_S_2_ nanowires cores are uniformly covered by Co(OH)_2_ nanosheets shells. The interphase between Ni_3_S_2_ and Co(OH)_2_ is no clear boundary observed, indicating that a close contact was formed between them. Such a structure may result in improved electrochemical performance as it favors electron transfer during the Faradaic redox reactions [45]. The diameters of the core Ni_3_S_2_ nanowires are ~150 nm, and the Co(OH)_2_ layers covering the Ni_3_S_2_ nanowires exhibit thicknesses in the range 20–40 nm. As shown in Figure 4c, the Co_0.5_–Ni_3_S_2_ nanowires exhibit uniform distributions of Ni, S, and Co. This further confirming the formation of a uniform, well-defined core–shell structure. The HRTEM image of the Co_0.5_–Ni_3_S_2_ nanowire (Figure 4d) reveals lattice spacings of 0.4, 0.28, and 0.24 nm, corresponding to the (110) and (101) interplanar spacings of Ni_3_S_2_ and the (101) interplanar spacings of Co(OH)_2_, respectively. As shown in the selected area electron diffraction (SAED) pattern (Figure 4e), several distinct concentric rings clearly indicate the polycrystallinity of Ni_3_S_2_, and the diffraction rings from inside to outside correspond to the (101), (110), (003), (202), (113), and (104) crystalline planes of Ni_3_S_2_, respectively.

Additional evidence regarding the chemical state of the as-synthesized Co_0.5_–Ni_3_S_2_ electrode was obtained using XPS. The full survey spectrum (Figure 5a) of the Co_0.5_–Ni_3_S_2_ electrode reveals that only Ni, Co, O, and C are present, with no other impurity peaks observed. The presence of Co and O is likely caused by the adsorption of Co(OH)_2_ at the surface of the Ni_3_S_2_ nanowire, and the small C presence may be due to the reference and the exposure of the sample to the atmosphere [46]. The high-resolution spectrum of Ni 2p (Figure 5b) shows two main core levels of Ni 2p_3/2_ and Ni 2p_1/2_ that are characteristic of the Ni state are located at 855.1 and 872.8 eV, with two satellite peaks at 860.9 and 878.9 eV, respectively, indicating the formation of Ni^2+^ and Ni^3+^ [47,48,49]. To investigate the Co(OH)_2_ shell formation on the surfaces of the Ni_3_S_2_ nanowires, the Co 2p XP spectrum was obtained, as shown in Figure 5c. Peaks representing Co 2p_3/2_ (780.4 and 774.0 eV) and Co 2p_1/2_ (796.5 and 796.0 eV) are observed. These peaks (796.0 and 780.4 eV) are attributed to Co^2+^, indicating the formation of Co(OH)_2_ [50,51,52]. For detailed insight into the presence of hydroxide, the O 1s spectrum of the Co_0.5_–Ni_3_S_2_ electrode was analyzed, as shown in Figure 5d. The curve fitting of the O 1s spectrum indicates that the sample consists of two components, represented by peaks at 531.5 and 530.4 eV. The 531.5 eV peak indicates a bound hydroxide group (OH^−^), confirming that Co(OH)_2_ is successfully formed on the Ni_3_S_2_ surface [53,54]. The small peak at 530.4 eV indicates the H–O bonds in water [55,56,57]. Therefore, the XP spectra, when combined with the XRD and TEM results, indicate the successful preparation of the Co(OH)_2_ shells on the surfaces of the Ni_3_S_2_ nanowire cores.

The electrochemical behaviors of the Ni_3_S_2_ and Co(OH)_2_–Ni_3_S_2_ electrodes were initially investigated using CV (Figure 6a). For Ni_3_S_2_, redox peaks are observed at 0.20 and 0.30 V, respectively, indicating pseudocapacitive behavior [58]. Compared to those of Ni_3_S_2_, the redox peaks shift negatively in the voltammogram of Co(OH)_2_–Ni_3_S_2_, indicating that the surface redox reactions of Co^2+^/Co^3+^/Co^4+^ and Ni^2+^/Ni^3+^ may facilitate OH^−^ transfer from the electrolyte [59,60]. The voltammogram of Co_0.5_–Ni_3_S_2_ clearly shows a large integrated area, suggesting that the former exhibits a higher electrochemical capacitance. Figure 6b shows the CV of the Co_0.5_–Ni_3_S_2_ electrode at scan rates ranging from 5 to 50 mV s^−1^. The response current density increases linearly with increasing scan rate and the cathodic and cathodic current peaks are shifted more positively and negatively, respectively. However, as the scan rates increase, clear redox peaks are observed, indicating the slow electron or ion transfer kinetics of the interfacial redox reactions [61]. Conversely, the voltammograms of Co_0.5_–Ni_3_S_2_ at scan rates of ≤20 mV s^−1^ retain their original shapes, and the currents are higher than that of Ni_3_S_2_, indicating the rapid kinetics of electron or ion transfer in the interfacial redox reactions [62]. Figure 6c shows the galvanostatic charge/discharge (GCD) curves of Co_0.5_–Ni_3_S_2_ at different current densities. All GCD curves shows the clear voltage plateaus suggesting Faradaic redox reactions. The specific capacitance, SCs, (C_S_, F g^−1^) and area capacitances (C_A_, F cm^−2^) of the electrodes are calculated based on the GCD curves using the following equations [63]:(1)$$C_{S}=\frac{I\cdot (\frac{\Delta t}{\Delta V})}{m}$$
(2)$$C_{A}=\frac{I\cdot (\frac{\Delta t}{\Delta V})}{A}$$
where I, ΔV, Δt, *m*, and A denote the discharge current (mA), working potential window (V), discharging time (s), deposited mass of the active material (g), and area of the NF coated with the active material (1 × 1 cm^2^), respectively. 

Figure 6d,e shows the rate capabilities of various electrodes calculated during GCD. Co_0.5_–Ni_3_S_2_ exhibits a very high SC compared to those of the other samples. The SC (plot with slid lines) is 742.6 F g^−1^ (9.25 F cm^−2^) at a current density of 10 mA cm^−2^, which declines to 451.8 F g^−1^ (5.63 F cm^−2^) when the current density increases to 50 mA cm^−2^. As shown in the GCD curve, the electrode materials of the Ni_3_S_2_ and Co-Ni_3_S_2_ shows the clearly redox plateau; therefore, the specific and area capacities (dotted lines) were also plotted in the Figure 6d,e. The discharge capacity of Co_0.5_–Ni_3_S_2_ shows that the 90.1 mAh g^−1^ at a current density of 10 mA cm^−2^ and 52.8 mAh g^−1^ at 50 mA cm^−2^. The high SC and capacity of Co_0.5_–Ni_3_S_2_ may be due to the unique core–shell nanowires, wherein highly conductive Ni_3_S_2_ with a high SC is critical in enhancing the capacitance of Co(OH)_2_ with an ultrathin nanosheet morphology. Figure 6f shows the cycling performances of Ni_3_S_2_ and Co–Ni_3_S_2_. Ni_3_S_2_ and Co_0.5_–Ni_3_S_2_ showed retention rates of 83.7% and 95.4%, respectively, up to 1000 cycles, indicating that the core–shell structure improves the long–term electrochemical stability of the KOH electrolyte system. The optimal SC and cycling performance of the electroactive Ni_3_S_2_ electrode are superior or comparable to those of previously reported Ni_3_S_2_ electrodes (Table 1) [25,64,65,66,67,68].

The electrochemical performance of the AC electrode in the three–electrode configuration was measured, with the AC coated on a nickel mesh (1 × 1 cm^2^) used as the working electrode. Figure 7 shows the electrochemical behavior of the AC electrode at different scan rates. The CV curves (Figure 7a) of the AC electrode reveal excellent symmetrical rectangular shapes at all scan rates. This is characteristic of an ideal EDLC material and suggests good charge propagation, rapid ion diffusion within the pores and reorganization of the electrical double layer, and low contact resistance [69]. Figure 7b,c show the GCD curves of the AC electrode at different current densities (10–50 mA cm^−2^) and the rate capabilities, respectively. The charge–discharge curves of AC electrodes are linear and symmetrical, indicating the good reversibility. The SCs of the AC electrode are calculated based on the GCD curves using Equation (1). An SC is 109.8 F g^−1^ (0.4 F cm^−2^) appears at 5 mA cm^−2^, which declines to 101.5 F g^−1^ (0.34 F cm^−2^) at 50 mA cm^−2^. The AC electrode maintains 99.5% of its initial capacitance after 2000 cycles as shown in Figure 6d.

The ASC device was assembled in a two-electrode configuration using a Co_0.5_–Ni_3_S_2_ positive electrode and an AC negative electrode. Prior to assembling the ASC device, the mass balance of the positive and negative electrodes was determined. The mass (m) of the positive and negative electrodes was determined by Equation (3) [70]:(3)$$\frac{m_{+}}{m_{-}}=\frac{C_{-}\times \Delta V_{-}}{C_{+}\times \Delta V_{+}}$$
where *m*, *C*, and Δ*V* are the masses, SCs, and potential windows of the Co_0.5_–Ni_3_S_2_ positive (+) and AC negative (−) electrodes, in a typical three–electrode configuration. The mass ratio of AC and Co_0.5_–Ni_3_S_2_ is 2.73:1. The SC of the device was calculated using Equation (1), where m is the sum of the deposited masses of Co_0.5_–Ni_3_S_2_ and AC (cathode and anode materials, respectively). The total active mass is 27 mg. The potential window of the Co_0.5_–Ni_3_S_2_//AC device is obtained from the cyclic voltammograms of Co_0.5_–Ni_3_S_2_ (Figure 6b) and AC (Figure 7a) in a three–electrode cell. AC exhibits the typical EDLC pattern in the potential range from −1.0–0.0 V and Co_0.5_–Ni_3_S_2_ exhibits the potential range 0–0.4 V. Based on the cyclic voltammograms of the Co_0.5_–Ni_3_S_2_ and AC electrodes, the potential window of the ASC device may be extended to 1.6 V [71].

The cyclic voltammograms exhibit large current areas with broad redox peaks, which are typical of asymmetric supercapacitors that combine electrochemical capacitors and EDLC properties [72]. The potential window of the Co_0.5_–Ni_3_S_2_//AC device was again confirmed using CV at different windows. As shown in Figure 8a, no clear polarization is observed in the voltage range 0.0–1.60 V, indicating that 1.40 V is a reasonable potential window limit for this ASC in 3.0 M KOH electrolyte. Therefore, the combined working potential of the ASC is 1.6 V. Figure 8b shows the CV of the ASC at different scan rates in the potential window 0–1.6 V. The CV exhibit broad redox peaks, which are associated with the pseudocapacitive Ni_3_S_2_ and AC, indicating the ideal capacitive behavior of the device [73]. Figure 8c shows the typical GCD curves of the ASC with observed SCs of 2.37 F cm^−2^ (58.41 F g^−1^) at 20 mA cm^−2^. In addition, the SC retention of the ASC device is 94% at 20 mA cm^−2^, as shown in the inset of Figure 8c. To study the interfacial charge transfer resistance of the supercapacitor with the Co_0.5_–Ni_3_S_2_//AC device, EIS was performed after the 1st and 1000th discharges, as shown in Figure 8d. The Nyquist plot in the high–frequency region reveals that the electronic resistance of the device, also denoted the solution resistance (R_S_), is the equivalent series resistance (ESR), which describes the resistance of the electrolyte combined with the internal resistance of the electrode [74]. The semicircle is the result of the double-layer charging of the AC electrode and the Faradaic reaction of the Co_0.5_–Ni_3_S_2_ electrode. The slope of the curve in the low-frequency region corresponds to the Warburg element, which represents the electrolyte diffusion within the porous electrode and proton diffusion within the host material [75,76]. The Nyquist plot was fitted using the Randles equivalent circuit, as shown in the inset of Figure 8d, using the NOVA program, where R_ct_ is the electrode-electrolyte charge transfer resistance, R_L_ is the leakage resistance, C_L_ is the double–layer capacitance, CPE_DL_ is the constant phase element of the double layer, and CPE_L_ is the mass capacitance. The ESR of the asymmetric device decreases from 0.25 to 0.22 Ω after 1000 cycles, and R_ct_ also decreases after 1000 cycles, suggesting a significant decrease in the R_S_ and R_ct_ due to the increased surface area available during repeated cycling, which may improve the supercapacitor performance [77].

The Co_0.5_–Ni_3_S_2_//AC asymmetric device exhibits a maximum energy density of 60.3 W h kg^−1^ (20 mA cm^−2^) at a power density of 1944.3 W kg^−1^ (50 mA cm^−2^) and 36.0 W h kg^−1^ at a power density of 4309 W kg^−1^. The energy (*E*, W h kg^−1^) and power densities (*P*, W kg^−1^) are calculated based on the discharge curve using Equations (4) and (5), respectively [78]:(4)$$E=\frac{I\int Vdt}{m\times 3.6}$$
(5)$$P=\frac{E\times 3600}{t}$$
where, *I*, ∫Vdt, *m*, and *t* are the discharge current (mA), integral area under the discharge curve, masses, and discharging time (s), respectively. The results are comparable or even superior to those of previously reported supercapacitors (Table 2), such as the rGO@Ni_3_S_2_//AC (37.19 W h kg^−1^ at 399.9 W kg^−1^), Ni_3_S_2_//AC (36 W h kg^−1^ at 400 W kg^−1^), and Ni_3_S_2_@CoS//AC (23.69 W h kg^−1^ at 268.95 W kg^−1^) devices [66,79,80,81,82,83,84,85]. A comparison of these results based on power and energy densities with related reported devices is also summarized. To further evaluate the practical applications of the Co_0.5_–Ni_3_S_2_//AC asymmetric device, the assembled device easily powers an LED after charging, as shown in the inset of Figure 8c. The Co_0.5_–Ni_3_S_2_//AC asymmetric device exhibits considerable potential for use in next–generation flexible energy storage devices, for applications in smart, wearable electronics.

## 4. Conclusions

Ni_3_S_2_ and Co(OH)_2_ shell-coated Ni_3_S_2_ nanowires were successfully prepared using a facile one–step hydrothermal reaction as positive electrodes for use in supercapacitors. The Ni_3_S_2_ nanowires were uniformly formed on the smooth NF surface, and the Co(OH)_2_ shell was uniformly formed on the Ni_3_S_2_ nanowire surface. As NF participates directly as the nickel source, Ni_3_S_2_ was formed more closely to the NF surface, and the Co(OH)_2_ shell suppressed the loss of active material during charging–discharging. The 0.5 mmol Co(OH)_2_ shell–coated Co_0.5_–Ni_3_S_2_ electrode exhibited superior performance, with a high discharge capacity of 1837 F g^−1^ (16.07 F cm^−2^) at a current density of 5 mA cm^−2^, and maintained a satisfactory rate capability of 583 F g^−1^ (16.07 F cm^−2^) at a much higher current density of 50 mA cm^−2^. The ASC delivers a power density as high as 1036 W kg^−1^ at an energy density of 43 W h kg^−1^, with a good cycling stability of 92%, which reveals its suitability for use in numerous energy storage applications. Therefore, the core–shell architecture developed in this study is an efficient way to improve the electrochemical performances in various promising energy storage applications.

## Figures and Tables

**Figure 1 nanomaterials-12-00034-f001:**
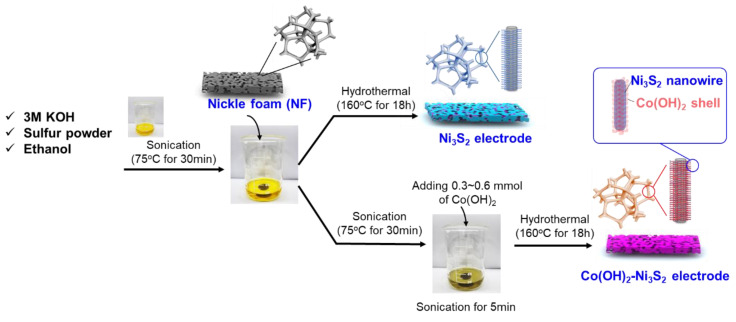
Schematic diagram of hydrothermal Ni_3_S_2_ and Co(OH)_2_–Ni_3_S_2_ production.

**Figure 2 nanomaterials-12-00034-f002:**
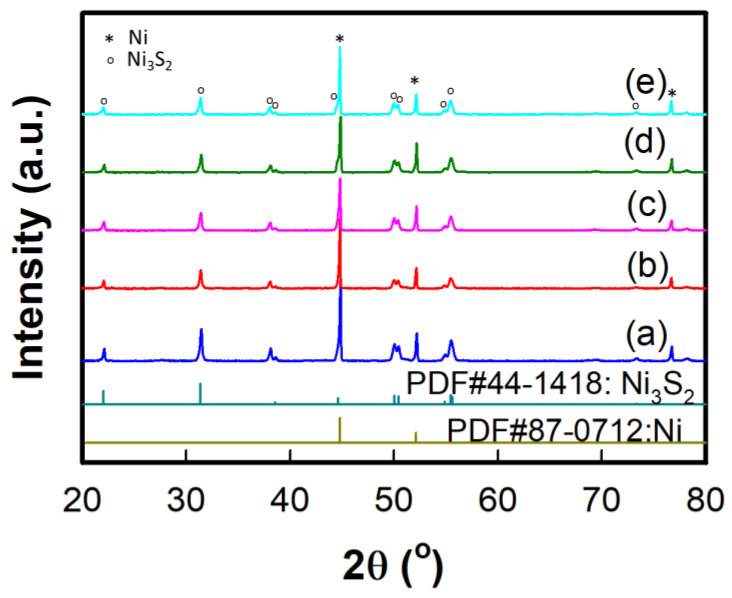
XRD patterns of the (**a**) N_3_S_2_, (**b**) Co_0.3_–Ni_3_S_2_, (**c**) Co_0.4_–Ni_3_S_2_, (**d**) Co_0.5_–Ni_3_S_2_, and (**e**) Co_0.6_–Ni_3_S_2_ electrodes.

**Figure 3 nanomaterials-12-00034-f003:**
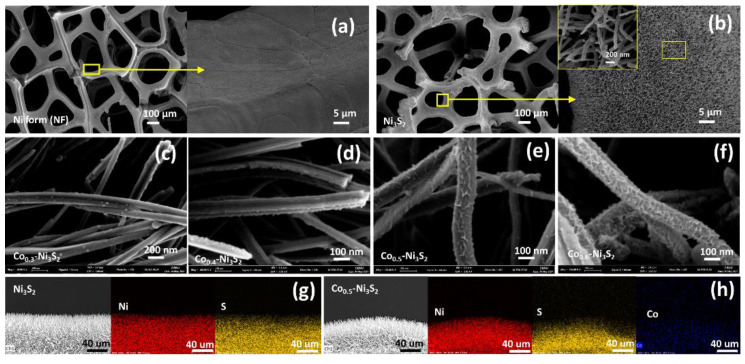
FE-SEM images at different magnifications and EDS mapping of (**a**) NF, (**b**) Ni_3_S_2_, (**c**) Co_0.3_–Ni_3_S_2_, (**d**) Co_0.4_–Ni_3_S_2_, (**e**) Co_0.6_–Ni_3_S_2_, (**f**) Co_0.6_–Ni_3_S_2_, (**g**) Ni_3_S_2_, and (**h**) Co_0.5_–Ni_3_S_2_.

**Figure 4 nanomaterials-12-00034-f004:**
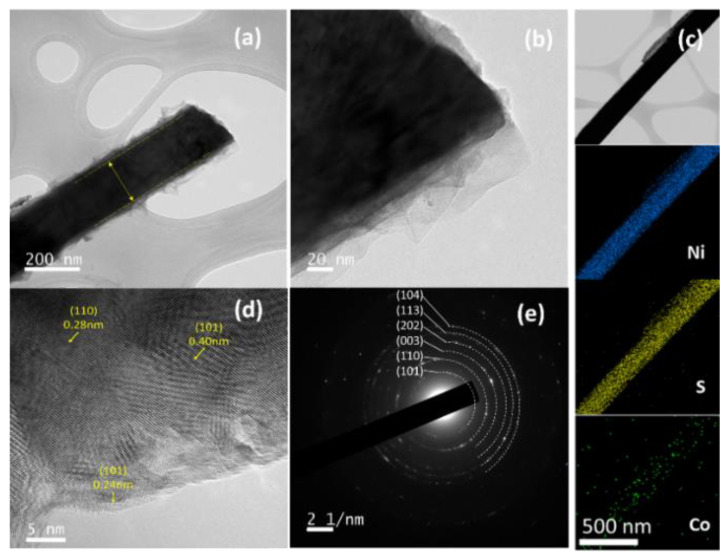
(**a**) Low and (**b**) high-magnification TEM images of the Co_0.5_–Ni_3_S_2_ nanowire, (**c**) EDS mapping images, (**d**) lattice-resolved HRTEM image of the Co_0.5_–Ni_3_S_2_ nanowire, and (**e**) SAED pattern of the Co_0.5_–Ni_3_S_2_ nanowire.

**Figure 5 nanomaterials-12-00034-f005:**
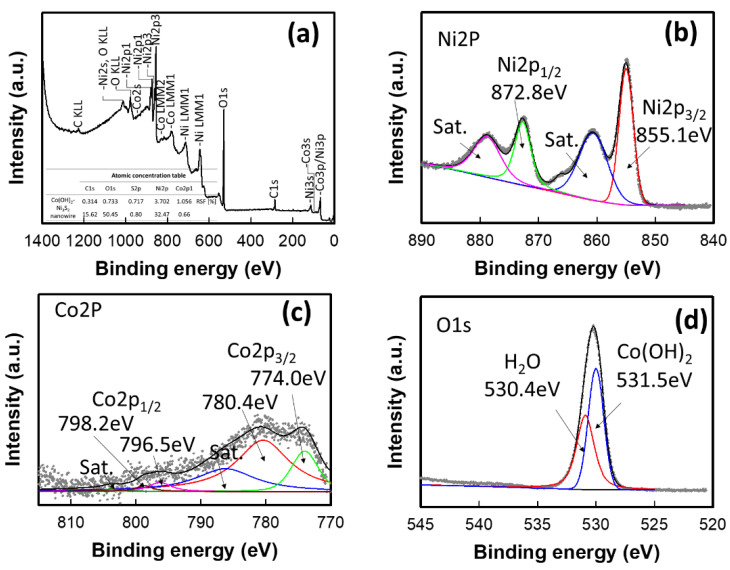
XPS spectra of the Co_0.5_–Ni_3_S_2_ electrode. (**a**) Survey, (**b**) Ni 2p, (**c**) Co 2p, and (**d**) O 1s electron XPS spectra.

**Figure 6 nanomaterials-12-00034-f006:**
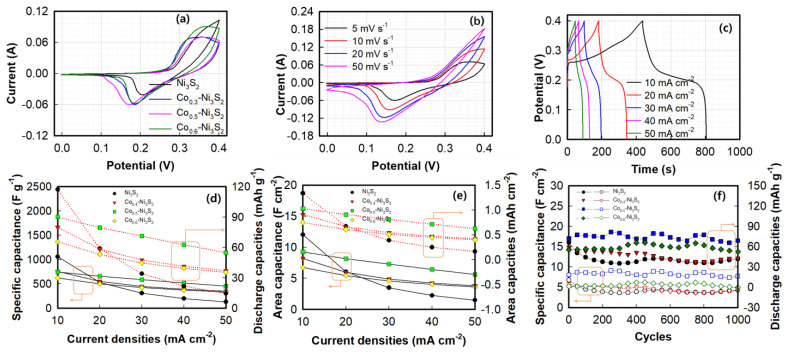
Cyclic voltammograms of (**a**) different electrodes at 5 mV s^−1^, (**b**) Co_0.5_–Ni_3_S_2_ at different scan rates, (**c**) galvanostatic charge–discharge of Co_0.5_–Ni_3_S_2_ electrode, (**d**) specific capacitances and capacities, (**e**) area capacitances and capacities, and (**f**) cycling performances at 20 mA cm^−2^.

**Figure 7 nanomaterials-12-00034-f007:**
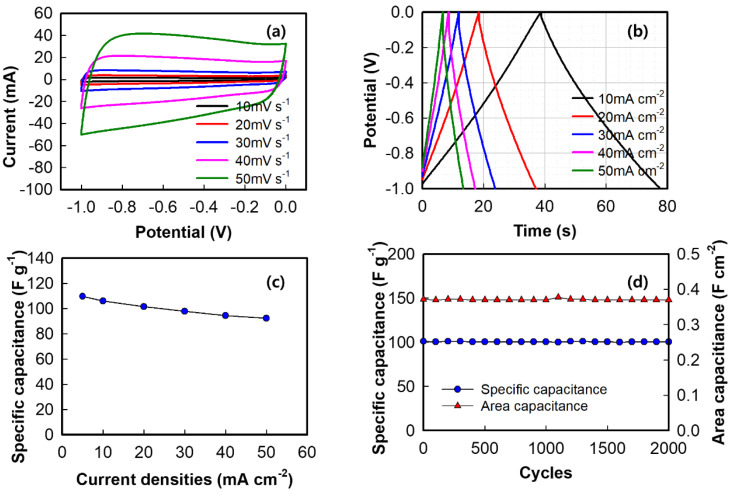
(**a**) Cyclic voltammograms of the AC negative electrode at 10–50 mV s^−1^, (**b**) galvanostatic charge–discharge curves of the AC electrode at 10–50 mA cm^−2^, (**c**) specific capacitances at different current densities, and (**d**) cycling performance of the AC electrode at 20 mA cm^−2^.

**Figure 8 nanomaterials-12-00034-f008:**
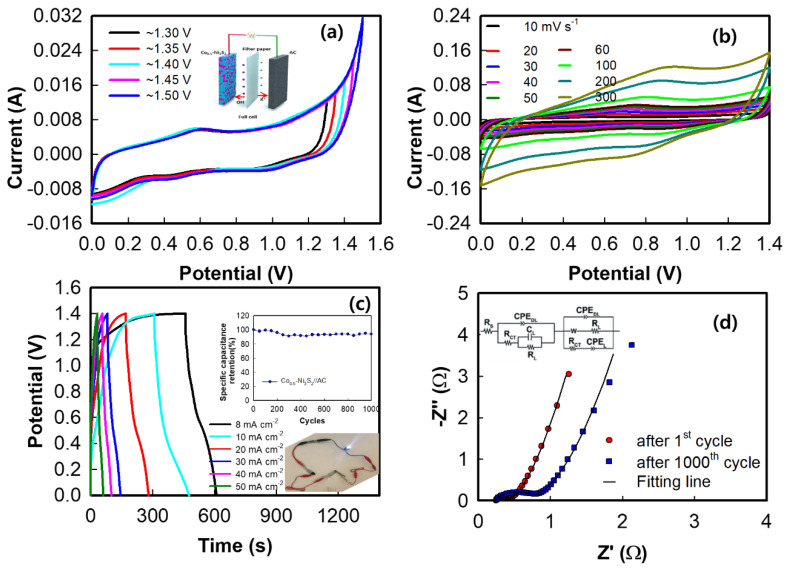
Cyclic voltammograms of the Co_0.5_–Ni_3_S_2_//AC device (**a**) in different voltage windows and (**b**) at different scan rates. (**c**) Galvanostatic charge–discharge curves of the Co_0.5_–Ni_3_S_2_//AC device at 8–50 mA cm^−2^ and (inset) cycle retention at 20 mA cm^−2^, and (**d**) EIS and fitting results of the Co_0.5_–Ni_3_S_2_//AC device after 1 and 1000 cycles.

**Table 1 nanomaterials-12-00034-t001:** Comparison of the electrochemical energy storage performances observed in this study with those of previously reported Ni_3_S_2_ electrodes.

Electrodes	Current Density	Specific or Areal Capacitance	Cycle Stability	Ref.
Ni_3_S_2_	15 A g^−1^	670 F g^−1^	97.4% (2000)	[50]
Ni_3_S_2_ nanosheet @ Ni_3_S_2_ nanorods	20.6 A g^−1^	489 F g^−1^	89.3% (5000)	[51]
Ni_3_S_2_ nanoflake	50 A cm^−2^	2.28 F cm^−2^	77% (5000)	[52]
Ni_3_S_2_ nanoporous	1 mA cm^−2^	3.42 F cm^−2^	102% (4250)	[53]
Ni_3_S_2_ nanosheet	15 mA cm^−2^	1.342 F cm^−2^	93.6% (3000)	[54]
Ni_3_S_2_ nest	25 mA cm^−2^	682.9 F g^−1^	69% (1000)	[55]
Ni_3_S_2_ nanowire	20 mA cm^−2^	530.3 F g^−1^ (6.05 F cm^−2^)	83.7% (1000)	This study
Co(OH)_2_–Ni_3_S_2_ nanowire	20 mA cm^−2^	656.2 F g^−1^ (8.17 F cm^−2^)	95.4% (1000)	

**Table 2 nanomaterials-12-00034-t002:** Comparison of the electrochemical energy storage performance observed in this study with those of previously reported asymmetric devices based on Ni_3_S_2_ and AC electrodes [66,79,80,81,82,83,84,85].

Electrodes	Energy Density (W h kg^−1^)	Power Density (W kg^−1^)	Ref.
rGO–Ni_3_S_2_//AC	37.19	399.9	[66]
Ni_3_S_2_//AC	36	400	[79]
Ni_3_S_2_@CoS//AC	23.69	268.95	[80]
Ni_3_S_2_//AC	10.01	150.12	[80]
Ni_3_S_2_/MWCNT–NC//AC	19.8	798	[81]
C@MnNiCo–OH/Ni_3_S_2_/NF//AC	36,000	799.95	[82]
Ni_3_S_2_@PPy//AC	17.54	179.33	[83]
Ni_3_S_2_//AC	17.73	179.32	[83]
NiSe/Ni_3_S_2_//AC	38.7	192	[84]
(Ni_3_S_2_/Ni@CC//AC/CC	0.27 W h cm^−2^	4.90 W cm^−2^	[85]
Co(OH)_2_–Ni_3_S_2_ nanowire	60.3	1944.3	This study

## Data Availability

Not applicable.

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
