# Peer review of "Facile In Situ Synthesis of Co(OH)2–Ni3S2 Nanowires on Ni Foam for Use in High-Energy-Density Supercapacitors"

_nanomaterials, 2021, doi:10.3390/nano12010034_

Round 1

Reviewer 1 Report

  1. Reactions for the formations of Ni3S2 and Co(OH)2 need to provide in the manuscript. Why does it form Ni3S2, not Co3S2? Also, why does it forms Co(OH)2, not Ni(OH)2? The reasons need to be provided in the manuscript.
  2. The mass loading of Co(OH)2 should be checked and provided. Its effect on the electrochemical performance should be discussed in the text. 
  3. The elemental percentage from the XPS analysis needs to be shown.
  4. Fig.8c shows the large IR drop and very low Coulombic efficiency of the ASC device at the potential widow of 0-1.4 eV. The authors may check at a smaller potential window.
  5. For practical application, more cyclic tests of the ASC device need to check and added at least 5000 cycles.

Author Response

  1. Reactions for the formations of Ni3S2 and Co(OH)2 need to provide in the manuscript. Why does it form Ni3S2, not Co3S2? Also, why does it forms Co(OH)2, not Ni(OH)2? The reasons need to be provided in the manuscript.
    Answer) Thank you for your comments. In the revised manuscript, the reactions for the formations of Ni3S2 and Co(OH)2 were added and the discussion was given in the paper.
    The reaction formular for the formation of nickel sulfide on the NF can be summarized as below;
    3OH- + 4S → 2S2- + S2O32- + 3H+
    Ni + 2OH- → Ni(OH)2 + 2e-
    3Ni(OH)2 + 2S2- → Ni3S2 + 6OH-
    And, for preparation of Co(OH)2 on the NF surface, after sonicating nickel at 70 oC was completed, a Co source was added. Therefore, in the hydrothermal reaction Ni(OH)2 reacts with S2- to foam NS, and the Co2+ reacts with OH- to form Co(OH)2.

  2. The mass loading of Co(OH)2 should be checked and provided. Its effect on the electrochemical performance should be discussed in the text.
  3. Answer) Thank you for your comments. The 0.3, 0.4, 0.5, or 0.6 mmol cobalt-containing Co(OH)2–Ni3S2 electrodes are denoted as Co3–, Co0.4–, Co0.5–, and Co0.6–Ni3S2 with a mass loading of 6.2, 6.8, 7.3 and 8.1 mg cm-2, respectively.
    “A rough surface with a large specific surface area may exhibit excellent electrolyte adsorption properties, and Co(OH)2 may suppress the Ni3S2 active material loss during the cycling, resulting in excellent electrochemical properties.”

  4. The elemental percentage from the XPS analysis needs to be shown.
    Answer) Thank you for your comments. The element percentage were provided in Figure 5(a) in the revised manuscript.

  5. 8c shows the large IR drop and very low Coulombic efficiency of the ASC device at the potential widow of 0-1.4 eV. The authors may check at a smaller potential window.
    Answer) Thank you for your comment. As shown in following GCD Figure (c), IR drop appeared at low current densities such as 8 and 10 mA cm-2, however IR drop phenomenon was improved at high current densities. Thus, in our study, the cycling was tested at 20 mA cm-2 and shows the cycle retention of 94%.

  6. For practical application, more cyclic tests of the ASC device need to check and added at least 5000 cycles.
    Answer) Thank you very much for your suggestion. As shown in the following supercapacitor related papers and there are many tests results up to 1000 cycles. Therefore, in this study, the cycling analysis was also performed up to 1000 cycles, but in our next study, we will publish results for more than 1000 cycles by reflecting the reviewer’s opinions.
    (1) Lee, C.-Y.; Yeh, F.-H.; Yu, I.-S. A Commercial Carbonaceous Anode with a-Si Layers by Plasma Enhanced Chemical Vapor Deposition for Lithium Ion Batteries. Journal of Composites Science 2020, 4, 72, doi:10.3390/jcs4020072.
    (2) Cheng, B.; Cheng, R.; Tan, F.; Liu, X.; Huo, J.; Yue, G. Highly Efficient Quasi-Solid-State Asymmetric Supercapacitors Based on MoS2/MWCNT and PANI/MWCNT Composite Electrodes. Nanoscale Res Lett 2019, 14, 66, doi:10.1186/s11671-019-2902-5.
    (3) Lai, F.; Fang, Z.; Cao, L.; Li, W.; Lin, Z.; Zhang, P. Self-healing flexible and strong hydrogel nanocomposites based on polyaniline for supercapacitors. Ionics 2020, 26, 3015-3025, doi:10.1007/s11581-020-03438-3.
    (4) Neeraj, N.S.; Mordina, B.; Srivastava, A.K.; Mukhopadhyay, K.; Prasad, N.E. Impact of process conditions on the electrochemical performances of NiMoO4 nanorods and activated carbon based asymmetric supercapacitor. Applied Surface Science 2019, 473, 807-819, doi:10.1016/j.apsusc.2018.12.220.

Reviewer 2 Report

The authors propose an approach for synthesis of Co(OH)2-Ni3S2 nanowires on Ni foam for use in high-energy-density supercapacitors. The work is interesting. I would be happy to reassess the suitability of the work for publication should the authors address my comments. 

The introduction is good but doesn't include an overview of other form of metal nanowires (e.g., niobium nanowire supercapacitors). 

How many folds of enhancement in specific capacitance (C/m^2) is observed after "coating" the Ni foam with the nanoparticles?

On the Nyquist plot, it looks like after 1000 cycles, the diameter of the semi-circle is increases, although the ESR slightly decreases. How can we explain this increase?

Author Response

  1. The authors propose an approach for synthesis of Co(OH)2-Ni3S2 nanowires on Ni foam for use in high-energy-density supercapacitors. The work is interesting. I would be happy to reassess the suitability of the work for publication should the authors address my comments. 
    Answer) Thank you for your positive comment.

  2. The introduction is good but doesn't include an overview of other form of metal nanowires (e.g., niobium nanowire supercapacitors). 
    Answer) Thank you for your Since our manuscript focuses on the nickel sulfide and core-shell structure, therefore the introduction mainly discusses the preparation method, morphology, and electrochemical properties of nickel sulfide. Therefore, we do not include any mention of other metal nanowires.

  3. How many folds of enhancement in specific capacitance (C/m^2) is observed after "coating" the Ni foam with the nanoparticles?
    Answer) Thank you for your comment. After coating NS surface with Co(OH)2, the specific capacitance dose increases by 23% as shown in the Table 1. More importantly, the coating also improved rate capability and cycling stability. These contents and reasons are explained in the results and discussion section of the manuscript.

  4. On the Nyquist plot, it looks like after 1000 cycles, the diameter of the semi-circle is increases, although the ESR slightly decreases. How can we explain this increase?
    Answer) Thank you for your positive comment. This can be explained by the increase in Rct due to damage to the electrode according to charging and discharging, as described in many references.
    (1) Gul, H.; Shah, A.A.; Bilal, S. Achieving ultrahigh cycling stability and extended potential window for supercapacitors through asymmetric combination of conductive polymer nanocomposite and activated carbon. Polymers 2019, 11,1678. doi:10.3390/polym11101678.
    (2) Wu, J.; Yu, H.; Fan, L.; Luo, G.; Lin, J.; Huang, M. A simple and high-effective electrolyte mediated with p-phenylenediamine for supercapacitor. Journal of Materials Chemistry 2012, 22,19025. doi:10.1039/c2jm33856d.
    (3) Lu, L.; Xu, S.; An, J. Electrochemical performance of Mn3O4/G/CB composite materials for supercapacitors. International Journal of Electrochemical Science 2016, 11, 6287-6296. doi:10.20964/2016.07.40.

Round 2

Reviewer 1 Report

The authors have addressed my comments. I recommended publishing this manuscript in Nanomaterials.

Author Response

Thank you for your positive comment.

Reviewer 2 Report

Good work.

Author Response

Thank you for your positive comment.